# Adverse Childhood Events, Post-Traumatic Stress Disorder, Infectious Encephalopathies and Immune-Mediated Disease

**DOI:** 10.3390/healthcare10061127

**Published:** 2022-06-17

**Authors:** Robert C. Bransfield

**Affiliations:** 1Rutgers-Robert Wood Johnson Medical School, Piscataway, NJ 08854, USA; bransfield@comcast.net; Tel.: +1-732-741-3263; 2Hackensack Meridian School of Medicine, Red Bank, NJ 07701, USA

**Keywords:** intrusive symptoms, autoimmune diseases, inflammation, trauma, tick-borne disease, Lyme disease, child abuse, immune, chronic infection

## Abstract

Adverse Childhood Events (ACE), Post-Traumatic Stress Disorder (PTSD), and infectious encephalopathies are associated with immune-mediated diseases. Data supporting this are reviewed, and an integrated hypothesis is provided. All three can be associated with intrusive symptoms and temporal lobe pathology. ACE and PTSD are accompanied by an impaired mental capacity to differentiate external danger vs. safety. Infectious encephalopathies are accompanied by a failure of adaptive immunity and an impaired immune capacity to differentiate internal danger vs. safety. All three conditions are associated with impairments to differentiate danger vs. safety and adapt effectively. There are reciprocal interactions between ACE, PTSD, and infectious encephalopathies with accompanying persistent immune activation. This is associated with immune dysregulation, chronic hyperarousal, activation of the stress response, and impairments of the fear recognition and response neural circuits, hypothalamic–pituitary–adrenal axis, amygdala, and hippocampus. The pathophysiological processes can result in a broad spectrum of chronic neuropsychiatric and somatic symptoms and diseases. Understanding the psychodynamic, neurological, neuroimmune, inflammatory and autoimmune components of this interactive process expands the effective treatment opportunities.

## 1. Introduction

Adverse Childhood Events (ACE) frequently result in a developmental form of Post-Traumatic Stress Disorder (PTSD) that continues into adulthood [1]. ACEs are traumatic events occurring before the age of 18 and they include all types of childhood neglect and abuse. This neglect and abuse may include emotional neglect, physical neglect, and/or household substance use, mental illness, criminal behavior, incarceration, domestic violence, physical abuse, emotional abuse, sexual abuse, and/or parental separation or divorce. 

ACEs result in significantly increased rates of immune-mediated diseases in adulthood. This is based upon a number of findings. First hospitalizations for any autoimmune disease increased with an increasing number of ACEs compared with persons with no ACEs. Persons with ≥2 ACEs were at a 70% increased risk for hospitalizations who showed increased markers for inflammatory mediated conditions, T helper type 1 (Th1) cells, and at an 80% increased risk for autoimmune disease who showed markers for autoimmune-mediated conditions, T helper type 2 (Th2) cells. These individuals were also at a 100% increased risk for rheumatic diseases compared to individuals with no history of ACEs. The same study also demonstrated childhood traumatic stress increased the likelihood of hospitalization with a diagnosed autoimmune disease decades into adulthood [2]. 

Childhood adversities and adulthood traumas were independently associated with an immune marker for chronic inflammation, C-reactive protein (CRP). Those who had experienced both childhood adversity and adult trauma had higher levels of CRP than those with adulthood trauma alone. This demonstrated that childhood adversity and adulthood trauma are independently associated with elevated inflammation [3]. Being bullied in childhood and adolescence also predicted increased low-grade inflammation in adulthood with persistently increased levels of CRP [4]. ACEs are also associated with increased health risks in adulthood, including lung disease, heart disease, cancer, and diabetes [5]. 

While ACE and PTSD can be environmentally traumatic, infectious encephalopathies can be physiologically traumatic and many infectious diseases have demonstrated a causal effect on psychiatric and general medical symptoms. One model used for an example is Lyme/tick-borne diseases. These patients can experience a spectrum of chronic neuropsychiatric and general medical clinical findings associated with persistent immune activation [6,7,8,9,10]. In addition to other symptoms, infectious encephalopathies can cause intrusive symptoms, which is a core symptom in the pathophysiology of PTSD [11]. Infections that can cause intrusive symptoms to include Lyme/tick-borne diseases, Group A streptococcal infections, COVID-19, *Bartonella henselae*, Japanese B encephalitis virus, herpes simplex virus 1, Epstein–Barr virus, Mycoplasma, Hong Kong influenza and coxsackie virus [10,12,13,14,15,16]. Intrusive symptoms combined with the stressors involved in accessing care for Lyme/tick-borne disease, COVID-19, or other complex, chronic infectious diseases can further exacerbate pre-existing PTSD symptoms with the associated persistent immune activation [11,17]. 

## 2. Materials and Methods

The literature regarding the association between childhood trauma, childhood adverse events, trauma, post-traumatic stress disorder, infectious encephalopathies, immune physiology, immune pathophysiology, and treatment options was reviewed in PubMed, Google Scholar and in the author’s archives. Relevant articles were retrieved. The relevant contributors were defined, including trauma, Post-Traumatic Stress Disorder, intrusive symptoms, and Adverse Childhood Events. The pathophysiology was reviewed and discussed. These data were then combined to develop an integrated model, and a hypothesis to explain the findings was proposed. Based upon this model, treatment options were then discussed.

## 3. Results

### 3.1. Relevant Contributors

#### 3.1.1. Trauma

Traumatic events overwhelm a person’s usual method of coping that normally gives a sense of control, connection and meaning, thereby impairing their fundamental sense of safety, security, empowerment, and confidence that they can adequately differentiate danger from safety and thereby predict and contend with adversity. Not all trauma results in the development of Post-Traumatic Stress Disorder.

#### 3.1.2. Post-Traumatic Stress Disorder 

The concept of Post-Traumatic Stress Disorder (PTSD) was first developed to describe syndromes soldiers experienced following war-related trauma. The definition has since been improved upon in editions of the American Psychiatric Association Diagnostic and Statistical Manuals. Intrusive symptoms consist of the re-experiencing of events associated with the trauma repeatedly intruding. This result in three basic symptom clusters—hyperarousal (fight), avoidance (flight) and psychic numbing (fright). The chronic stress with PTSD is associated with impaired fear recognition and response neural circuits and hypothalamic–pituitary–adrenal axis functioning. The presence of PTSD results in a very high lifetime risk of major depressive episodes, alcohol abuse/dependence, drug abuse, social anxiety disorder, agoraphobia, generalized anxiety disorder, and panic disorder [18]. Patients with PTSD have difficulty differentiating safety from danger, particularly in circumstances associated with the prior trauma. PTSD persists as long as there is a failure to adapt to prior environmental trauma.

#### 3.1.3. Intrusive Symptoms

Intrusive symptoms are not voluntarily generated and are ego-dystonic. They can include images, thoughts, emotions, sensory perceptions, and/or re-experiencing of prior trauma. Intrusive symptoms are seen with PTSD, Obsessive-Compulsive Disorder, and infectious encephalopathies, such as Lyme/tick-borne diseases. Intrusive symptoms may be horrific, aggressive, sexual and/or bizarre. Intrusive symptoms at night can take the form of nightmares. The repetitive and unpredictable nature of intrusive symptoms can impair threat recognition and response and exacerbate PTSD symptoms. Only one article specifically addresses intrusive symptoms and infectious encephalopathies [11]. In addition, other articles on Lyme/tick-borne diseases include references to intrusive symptoms and infectious encephalopathies [10,16,19,20,21,22,23,24,25]. The mechanisms of infection-causing immune activation and thereby causing neuropsychiatric symptoms in Lyme/tick-borne diseases have been previously well described [6,7,8,9,26].

#### 3.1.4. Adverse Childhood Events

The concept of developmental trauma causing mental illnesses has long been recognized by Freud and others. Bessel van der Kolk has advanced the concept of developmental trauma disorder associated with Adverse Childhood Events [1]. ACE can be neglect, abuse and/or trauma. Neglect can be emotional, physical, medical, and/or educational. Abuse can be verbal, physical, and/or sexual [27]. Childhood trauma can include a parent’s divorce or separation, substance abuse in the home, death of a close family member, domestic abuse in the family, mental illness in the family, criminal behavior in the household and other traumatic events [28].

Adverse childhood events are associated with significant increases in the likelihood of depression, attempted suicide, suicide, sexual assault, domestic violence, alcoholism, drug abuse, intravenous drug abuse, lung disease, heart disease, cancer, diabetes, obesity, activation of latent chronic infections, and autoimmune diseases. The adverse effects of ACE can occur years and decades later [2].

### 3.2. Pathophysiology

Studies comparing different types of childhood and adult trauma demonstrate that a greater additive lifetime exposure to trauma exposure correlates with chronic stress and greater laboratory findings associated with immune activation with inflammation [29]. This persistent elevation of inflammation contributes to causing psychiatric symptoms as well as cardiovascular and autoimmune diseases [30,31,32,33,34,35]. One of the laboratory markers demonstrating persistent inflammation is C-reactive protein (CRP). Increases in the inflammatory marker, CRP, prior to the trauma are prospectively associated with a greater risk of developing PTSD from a traumatic event [2]. Adverse childhood events, as well as PTSD acquired in adulthood, are associated with increased levels of CRP [3,4]. In addition, individuals who experience clinical manifestations of PTSD exhibit higher CRP levels [36,37]. The development of clinical symptoms is also associated with increases in other inflammatory markers that are activated in the inflammatory cascade. These increased markers include Tumor Necrosis Factor Alpha and Interleuken-6 [38,39]. PTSD patients who experience intrusive symptoms also demonstrate increased inflammatory markers. In these patients, the severity of symptoms also correlates with activation of the basolateral amygdala and the ventral hippocampus and volumetric brain changes in these areas [40,41]. ACE is also associated with white matter microstructure disruption [42,43,44,45,46]. Aberrant reactivity of immune cell activity within the brain by microglia cells is another component of disease progression [47]. ACE has been associated with not only proinflammatory cytokines, but also a shorter telomer length, susceptibility to activation of latent infections, and impaired immune response to tumors [48]. Both ACE and PTSD have a negative lifelong impact on mental and physical health. They result in epigenetic modifications (such as methylation) and impaired functioning of the fear recognition and response neural circuits and hypothalamic–pituitary–adrenal (HPA) axis in the stress response system [49]. The reciprocal interaction between the brain and the immune system makes it possible for childhood psychosocial stressors to affect immune system development, which, in turn, can adversely affect brain development and its long-term functioning [50].

Lyme/tick-borne diseases and PTSD have a complex association. Inflammatory brain changes caused by the infection can cause intrusive symptoms [11,51]. However, some Lyme/tick-borne disease patients have comorbid PTSD from trauma unrelated to acquiring the infection. In addition, some patients develop PTSD as a result of trauma associated with having the infection. This trauma can include dealing with an overwhelming multisystem illness and inadequate support or rejection from family, friends, employers, doctors, government agencies, health insurers, disability insurers, and others who fail to recognize the severity of the condition. This trauma combined with the physiological effects of the infection may intensify the co-existing PTSD. 

While ACE and PTSD are associated with mostly external threats and stressors, Lyme/tick-borne diseases and other complex, chronic infections are a model of a complex interactive infection resulting in a pathological component to the microbiome causing an internal threat and stressor. Pathological microbiome activity is seen when there is a failure of adaptive immunity resulting in immune provocation with persistent inflammation and/or autoimmunity. The combination of direct microbial effects and immune provocation from these microbes results in symptoms [6,7,8,9]. These symptoms can cause disease progression and chronic stress. Chronic stress compromises restorative sleep and other restorative functions which impedes adaptive immunity and further contributes to disease progression [52]. 

Disease progression is frequently associated with an interaction of pathological microbiome activity and/or other environmental contributors, immune provocation, symptoms, a chronic stress response, compromised immune functioning and is further exacerbated by disease denial (Figure 1). 

## 4. Discussion

### 4.1. Overview

Adverse Childhood Events frequently result in a developmental form of Post-Traumatic Stress Disorder that continues into adulthood. Adverse Childhood Events, Post-Traumatic Stress Disorder acquired in adulthood, and some infectious encephalopathies can contribute to causing a similar complex of symptoms. This complex of symptoms includes intrusive symptoms, chronic stress, a failure to contend with environmental threats, a failure of adaptive immunity, chronic inflammation, autoimmunity, and impaired neuropsychiatric development. 

Intrusive symptoms are a significant component of these three conditions. This symptom is associated with disease perpetuation and progression and temporal lobe activation and pathology. 

There is also a reciprocal interaction between ACE, PTSD and chronic immune activation. ACE and PTSD cause a chronic hyperarousal state resulting in chronic activation of the stress response, impaired fear recognition and response neural circuits and impaired hypothalamic–pituitary–adrenal axis functioning, impaired amygdala and hippocampus functioning, and perpetuation of the inflammatory cytokine cascade. Furthermore, chronic immune activation evoked by chronic, complex interactive infections or other causes of chronic immune activation can result in chronic activation of the stress response, and a broad spectrum of chronic neuropsychiatric and somatic diseases. The chronic stress and hyperarousal associated with these three conditions is also associated with immune dysfunction consisting of a failure of adaptive immunity, persistent inflammation, and autoimmunity. The chronic stress response compromises regenerative functioning which results in poor overall health. 

Conceptually, the consequences of Adverse Childhood Events and Post-Traumatic Stress Disorder, persistent inflammation, and autoimmune disease result in maladaptive responses to danger. With the exception of PTSD associated with medical conditions, PTSD is mostly a maladaptive response to external danger, while persistent inflammation and autoimmune diseases are maladaptive responses to internal danger. Both PTSD and immune disorders can have a reciprocal intensification effect upon each other. 

Both ACE and PTSD acquired in adulthood are associated with an impaired capacity to differentiate environmental danger vs. safety. Chronic immune-mediated diseases are an impaired capacity to differentiate internal danger vs. safety. 

Chronic complex interactive infections can evoke a danger signal, but also have immune-evasive and immune-suppressive effects that result in a failure of adaptive immunity and perpetuation of chronic immune activation associated with chronic inflammation and/or autoimmunity [7]. 

ACE, PTSD and autoimmunity are all associated with maladaptive reactivity to cues in which danger and safety are incorrectly differentiated. Being in a chronic state of stress compromises regenerative functioning and increases susceptibility to multiple psychiatric and somatic illnesses (Figure 2).

### 4.2. Treatment Implications

A comprehensive assessment combined with clinical judgment can provide insight regarding which disease contributors are the most significant and the best sequence for treatment interventions by prioritizing which disease contributors are the most significant. Treatment can then be strategized to see where a well-tolerated treatment can have the most effective impact. Response or lack of response will then determine subsequent treatment strategies. Every patient is unique, and treatments always need to be individualized. Immune modulation strategies will depend upon whether there is a predominance of inflammatory (Th1) or autoimmune (Th2) symptoms. Anti-infective treatment is dependent upon whether a persistent infection is driving the immune response and which infection, or infections are present. Symptomatic treatment is focused on reducing the symptom(s) that contributes to the perpetuation and/or progression of symptoms. The goal in psychotherapy with PTSD is to regain a fundamental sense of safety, security, and empowerment with confidence in the ability to improve trust assessment capability and to better differentiate safety from danger. A number of psychotropic medications are used in the treatment of PTSD. Sertraline and paroxetine are approved by the United States Food and Drug Administration for the treatment of PTSD [53,54]. Prazosin and topiramate can also reduce intrusive symptoms [55,56]. 

Treatment options include interventions directed towards the contributors that drive disease perpetuation and progression. Therefore, effective treatments may include immune modulation, anti-infective treatments, symptomatic treatment (psychotherapy and psychotropic medications targeted towards treating PTSD and other significant symptoms) and education (Figure 3).

In summary, insight into the underlying pathophysiology of Adverse Childhood Events, Post-Traumatic Stress Disorder, and infectious encephalopathies offers expanded treatment opportunities that include psychotherapeutic treatments that help to differentiate safety vs. danger, therapies to improve empowerment, and psychotropic medications to reduce intrusive symptoms and other symptoms, anti-infective treatments, immune-modulating treatments and education for patients and caregivers.

## Figures and Tables

**Figure 1 healthcare-10-01127-f001:**
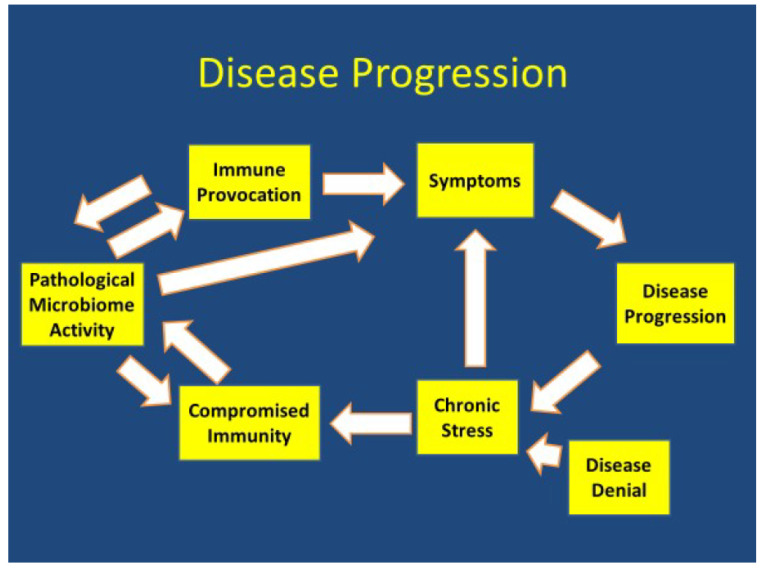
Disease Progression.

**Figure 2 healthcare-10-01127-f002:**
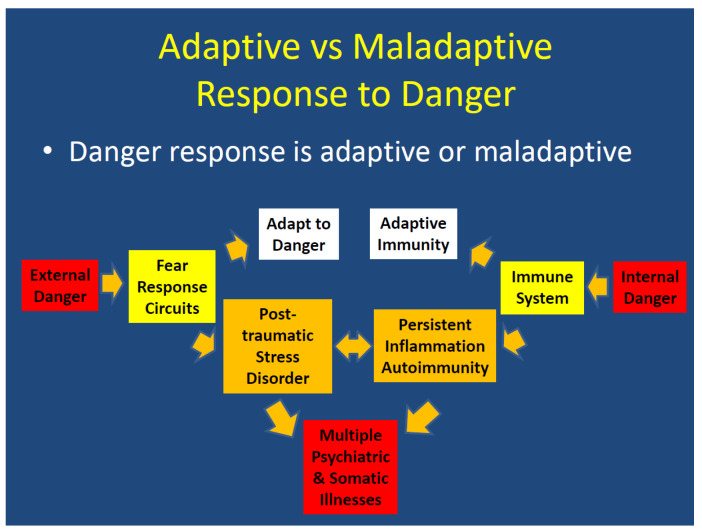
Adaptive vs. Maladaptive Response to Danger.

**Figure 3 healthcare-10-01127-f003:**
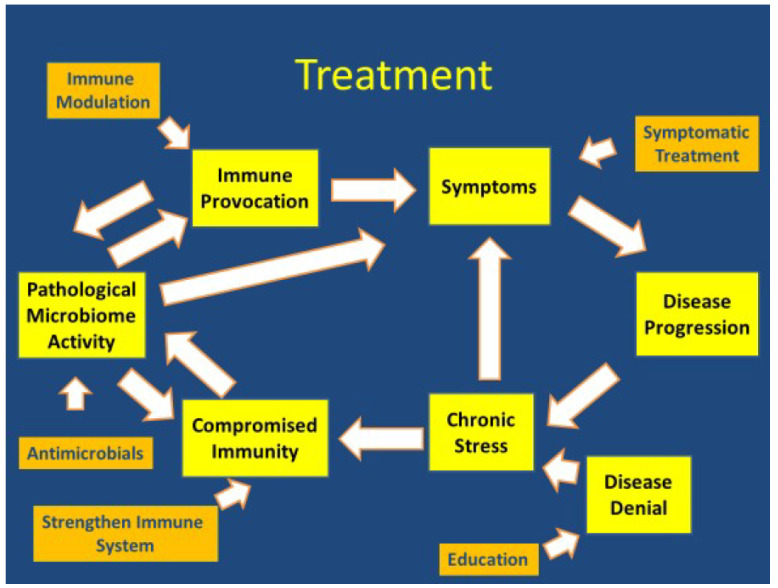
Treatment.

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
