# Peer review of "Adverse Childhood Events, Post-Traumatic Stress Disorder, Infectious Encephalopathies and Immune-Mediated Disease"

_healthcare, 2022, doi:10.3390/healthcare10061127_

Round 1

Reviewer 1 Report

Thank you for the opportunity to review, “Adverse Childhood Events, Posttraumatic Stress Disorder, In-

fectious Encephalopathies and Immune Mediated Disease.”

The review article is appropriate for Healthcare and provides a concise update for researchers and public health practitioners regarding ACEs. The broad topic includes numerous types of ACEs, including neglected areas such as infectious encephalopathies, with a focus on tick-borne diseases. These are topics deserving of attention and publication.  I recommend publication with minor changes.

Given Healthcare’s broad audience, not everyone reading this article will be familiar with some of the medical terminology. I suggest reviewing the manuscript for areas where non-medical experts/MDs may lack knowledge and provide a brief explanation.  I also suggest the following minor changes:

  • Define ACE broadly after the first sentence – establish the broad comparisons between childhood abuse, for example, and an infectious disease at the beginning of the paper. In the introduction, ACE is described as more of an umbrella term for all trauma or infectious disease type events. However, at the beginning of the results section, ACE is defined as a separate category, in addition to trauma or pathophysiology. The introduction should organize the terminology in manner that allows for a straightforward transition to results.  
  • Is there a citation that links Lyme or tick-borne disease to encephalitis? Explain the connection briefly for non-MDs, indicating why it falls under the ACE category (or if ACE is not an umbrella term, briefly indicate that upfront).
  • Lyme and tick-borne diseases are discussed throughout. However, other infectious agents are largely ignored. I would suggest an initial introduction to this topic and offer a reason for the focus on tbds/LD, e.g., these diseases are particularly complex with established inflammatory responses. They may also be more common than other infectious encephalopathies? e.g., rubella since there is a vaccine for many viruses that might lead to encephalitis but not for Lyme? I recommend a brief note regarding the focus on one over the others and/or introduce a few other examples caused by viruses or fungal agents to supplement the TBD discussion.

Author Response

Reviewer Response

Reviewer 1, Round 1

Thank you for your insightful comments. I did a number of revisions based upon your comments, and it has improved the overall flow of the article.

"Can be improved: Are the results clearly presented?"

 The Results section was better organized.

"The review article is appropriate for Healthcare and provides a concise update for researchers and public health practitioners regarding ACEs. The broad topic includes numerous types of ACEs, including neglected areas such as infectious encephalopathies, with a focus on tick-borne diseases. These are topics deserving of attention and publication.  I recommend publication with minor changes.

Given Healthcare’s broad audience, not everyone reading this article will be familiar with some of the medical terminology. I suggest reviewing the manuscript for areas where non-medical experts/MDs may lack knowledge and provide a brief explanation."

I went through the entire manuscript and changed wording to make it easier to understand for someone with limited scientific knowledge.

" I also suggest the following minor changes: Define ACE broadly after the first sentence"

Added after first sentence: ACEs are traumatic events occurring before the age of 18 and they include all types of childhood neglect and abuse. This neglect and abuse may include household substance use, mental illness, incarceration, domestic violence. physical/emotional/sexual abuse, and/or parental separation/divorce.

"– establish the broad comparisons between childhood abuse, for example, and an infectious disease at the beginning of the paper. In the introduction, ACE is described as more of an umbrella term for all trauma or infectious disease type events. However, at the beginning of the results section, ACE is defined as a separate category, in addition to trauma or pathophysiology. The introduction should organize the terminology in manner that allows for a straightforward transition to results."  

Wording was changed to make the distinctions clearer and to make better flow from Introduction, Method, Results, Discussion, and abstract

"Is there a citation that links Lyme or tick-borne disease to encephalitis? Explain the connection briefly for non-MDs, indicating why it falls under the ACE category (or if ACE is not an umbrella term, briefly indicate that upfront)."

Citations added in the Introduction.

"Lyme and tick-borne diseases are discussed throughout. However, other infectious agents are largely ignored. I would suggest an initial introduction to this topic and offer a reason for the focus on tbds/LD, e.g., these diseases are particularly complex with established inflammatory responses. They may also be more common than other infectious encephalopathies? e.g., rubella since there is a vaccine for many viruses that might lead to encephalitis but not for Lyme? I recommend a brief note regarding the focus on one over the others and/or introduce a few other examples caused by viruses or fungal agents to supplement the TBD discussion."

Although many infections are associated with psychiatric symptoms, there is little in the literature connecting intrusive symptoms to infections. I added the list of infections associated with intrusive symptoms.

Reviewer 2 Report

The review article provides a fair assessment of the impact of ACE and PTSD, however, a presentation of the magnitude of the impact of each which identifies the levels of CRP, etc. would improve the presented analyses.

Author Response

Reviewer 2, Round 1

Thank you for your attention and insightful comments. A number of revisions were made throughout the article to improve the overall flow of the article.

"Can be improved: Are the methods adequately described?"

               The entire article was organized better to match the description of the Method.

"Can be improved: Are the results clearly presented?"

            The Results section was better organized and made more consistent with the Method, Discussion and conclusions.

"Can be improved: Are the conclusions supported by the results?"  

The flow of the article was better organized to more clearly make the connection between results and conclusions.

"The review article provides a fair assessment of the impact of ACE and PTSD, however, a presentation of the magnitude of the impact of each which identifies the levels of CRP, etc. would improve the presented analyses."

I expanded and clarified the discussion on CRP. Due to the heterogeneity of the patients in these studies and the differences between the studies, it was difficult to do a valid analysis comparing the CPR levels in the different studies.

"I suggest reviewing the manuscript for areas where non-medical experts/MDs may lack knowledge and provide a brief explanation."

               This was done throughout the entire manuscript, and the wording was improved to make the manuscript easier to follow for non-medical experts. Although the content of this revision is basically the same, it now flows better and is easier to follow for someone with limited technical background. Every section was revised to improve clarity, including the abstract.